# Perceived Social Support Increases Creativity: Experimental Evidence

**DOI:** 10.3390/ijerph191811841

**Published:** 2022-09-19

**Authors:** Chee-Seng Tan, Xi-Yuan Chin, Samuel Ta-Chuan Chng, Jazen Lee, Chia-Sin Ooi

**Affiliations:** Department of Psychology and Counselling, Faculty of Arts and Social Science, Universiti Tunku Abdul Rahman (UTAR), Kampar 31900, Malaysia

**Keywords:** creativity, divergent thinking, experiment, perceived social support, Malaysia

## Abstract

The literature has consistently shown that social support has a positive relationship with creativity. However, further investigation is needed to clarify the causal relationship between the two constructs. The present study addressed this need by exploring the impact of experimentally induced perceived social support on creativity among young adults. A total of 135 undergraduate students in Malaysia participated in an online experiment. All participants first answered the creative self-efficacy scale and were then randomly allocated to the experimental and control groups. Perceived social support was primed by a writing test and measured by the Multidimensional Scales of Perceived Social Support. Both groups also answered a divergent thinking test (measured for fluency, flexibility, and originality) and a self-rated creativity scale. Multivariate analysis of covariance showed that, after statistically controlling for the effect of creative self-efficacy, participants in the experimental group reported higher scores in perceived social support and all creativity measures than their counterparts in the control group. The results demonstrated that the manipulation is effective and the induced perceived social support leads to higher creativity. Our findings not only offer empirical evidence of the causality of social support and creativity but also has practical value for creativity development.

## 1. Introduction

Creativity has been consistently recognized as a critical competency in the development and sustainability of organizations. Empirical studies have also shown that creativity is critical to individuals’ well-being and performance. For instance, creativity is negatively associated with behavioral and social instability [1] and stress [2], while positively correlated with well-being [3,4] and work performance [5]. Moreover, creativity has been found to enhance university students’ subjective well-being [6].

Creativity is “the interaction among aptitude, process and environment by which an individual or group produces a perceptible product that is both novel and useful as defined within a social context” [7] (p. 90). In other words, multiple sources are required for the development of creativity. Indeed, empirical findings from different disciplines have shown that creativity can be benefited by a variety of factors such as personality [8], autonomy [9], and culture, executive thinking style, and knowledge fixation [10], just to name a few. Similarly, social support was consistently found to have a positive correlation with creativity [11,12]. Notably, the beneficial relationship was documented in different contexts and populations, implying that social support is one of the key factors for creativity. The main objective of the present study is to extend the past correlational findings by shedding light on the causality of social support and creativity. In the next section, we briefly review the relationship between the two constructs, followed by an overview of the present study.

## 2. Perceived Social Support and Creativity

Social support refers to the availability of individuals that could provide potential or actual support (i.e., structural and functional support) [13] in terms of personal resources [14]. Social support has also been divided into two sub-categories: received social support and perceived social support [15]. Received social support is the actual amount of support acquired [16], while perceived social support is the personal belief or assessment of the degree to which social networks (e.g., family and friends) provide informational, physical, or psychological support [17]. The latter was found to play a more important role than received social support and, hence, has received more attention [18].

The literature has shown that perceived social support is positively related to happiness [19], hope [20], life satisfaction [21], resilience [22], post-traumatic growth [23], and physical health [14] and is negatively related to mental health challenges [24]. Specifically, the outbreak of COVID-19 highlighted the need for social support to elevate the mental health of individuals [16].

Social support is also positively associated with creativity. The positive relationship is theoretically supported by the self-determination theory (SDT) [25], which posits that autonomy, competence, and relatedness are the fundamental psychological needs of humans. Autonomy is the feeling individuals have when they can control their own lives; competence is the feeling individuals have when they can do something effectively, while relatedness refers to the need for a meaningful connection and relationship with others, including being part of a group in a social context. Social support can fulfil the three psychological needs [26], and the latter are useful for creativity. When people perceived that they were supported by their social network, they not only felt that they related to their supporters (i.e., high relatedness) but also felt competent to tackle the challenges and uncertainty in the process of generating new ideas. For example, when employees believed that their organization cared about them, they were autonomously motivated, which in turn benefited their creativity [9].

Empirical evidence for the positive association between social support and creativity was observed in the working contexts [27,28,29]. For instance, Talebzadeh and Karatepe [30] conducted a study on Iranian cabin attendants and discovered that supports from supervisor and coworkers played an important role in fostering employees’ creative performance. Similarly, a cross-sectional study conducted in South Korea discovered that perceived organizational support was beneficial to employees’ creativity when they experienced a high challenge stressor [31]. Moreover, a study that was conducted in Indonesia among small and medium enterprise owners also revealed that social support had a positive relationship with creativity and creative self-efficacy [32]. On top of organizational support, family social support was also found to have a positive relationship with the creativity of working adults [33].

Similarly, the social support–creativity association was also documented in educational contexts [34]. When examining the relationship between self-perceived creativity and entrepreneurial intentions among 559 students in Spain, Laguía et al. [35] found that students who perceived high levels of support from family and the university reported a higher level of creativity. Similarly, university students in groups that demonstrated supportive interactions were found to have better performance (i.e., fluency, flexibility, and originality scores rated by two independent coders) and higher self-rated creativity in idea generation and poster creation tasks, respectively [36].

## 3. The Present Study

Although there is substantial empirical evidence for the positive relationship between social support and creativity, most of the studies applied cross-sectional design, and hence, the causal relationship between the two constructs remains unclear. To address this gap, the present study employed the experimental design to manipulate perceived social support (i.e., individual’s belief of the readiness of psychological support from social relationships when needed) using a priming task and then examined the impact of the induced perceived social support on divergent thinking (indexed by fluency, flexibility, and originality), self-reported creativity, and creative self-efficacy, respectively. The findings of the study are important to the literature for two reasons. First, it would extend the literature by shedding light on causality. Second, it emphasizes the importance of perceived social support in enhancing creativity.

## 4. Method

### 4.1. Participants

According to the G-power estimation for F tests (for MANOVA: global effects) with a moderate effect size (f^2^ = 0.15), power of 0.95, and α = 0.05, a total of 146 participants were required to detect the treatment effect. At the end of the data collection period, 141 Malaysian undergraduates (73 females) were recruited using convenient sampling through different social media platforms (e.g., Facebook, Whatsapp). However, 5 participants who provided nonsensible responses to the divergent thinking test were removed from the further analysis, resulting in 135 participants (66 males and 69 females, *M*_age_ 22.01, *SD* = 1.15, range = 20 to 26). A post-hoc power analysis using G-Power (f^2^ = 0.35318) showed a power of 0.999.

Most of the participants identified themselves as Chinese (97.8%), followed by Malay (1.5%) and others (0.7%). The sample consisted of students from different disciplines ranging from social science (e.g., psychology, public relations), business (e.g., banking and finance, business administration), to science (e.g., biotechnology, electronic engineering). Data were collected through online meeting platforms, such as Microsoft Team and Google Meet, from February to March 2021. The students participated in the present study voluntarily.

### 4.2. Instrument

#### 4.2.1. Manipulation of Perceived Social Support

Following the practice of Xu and colleagues [37], perceived social support was manipulated using a writing task. Specifically, participants in the experimental group were asked to recall and write down three memories of being socially supported, while the control group was required to write down the three nearest objects around them.

#### 4.2.2. Multidimensional Scale of Perceived Social Support (MSPSS) [38]

The MSPSS was used to examine the effectiveness of the manipulation task. All participants responded to the 12 items using a 7-point Likert scale ranging from 1 (very strongly disagree) to 7 (very strongly agree). Higher scores indicate higher levels of perceived social support. A sample was “My friends really try to help me”. Cronbach’s alpha (α) was 0.860 while McDonald’s omega (ω) was 0.856.

#### 4.2.3. Alternative Uses Test

The Alternative Uses Test (AUT) [39] measures one’s divergent thinking capabilities. The participants were instructed to think of as many alternative uses for a book as possible within 3 min. The answers provided by each participant were then evaluated on three different dimensions: fluency, flexibility, and originality. Fluency is the total number of sensible answers generated by participants. On the other hand, flexibility is the number of different categories of sensible answers. Finally, originality refers to the extent to which the answer is uncommon compared to the other answers [40]. The originality of each answer was first rated on a 3-point scale ranging from 1 (low originality, i.e., the idea was generated by more than 50% of the participants), 2 (moderate originality, i.e., generated by 10% to 50%), to 3 (high originality, i.e., generated by below 10%). Then, the scores were summed up to generate a composite (originality) score for each participant.

The (second to fourth) researchers rated each participant’s responses for the three dimensions individually and separately. The three rating scores were averaged if they were highly consistent to represent participants’ performance in the three divergent thinking indexes.

#### 4.2.4. Self-Rated Creativity Scale (SRCS) [41]

The 12-item SRCS was used for participants to report their perceived creativity. Participants answered each item using a 5-point Likert scale ranging from 1 (*strongly disagree*) to 5 (*strongly agree*). The item scores were averaged to produce a composite score. Individuals who reported higher scores perceived themselves as more creative. A sample of the scale would be “I am a good source of creative ideas”. The SRCS showed good internal consistency (α = 0.829; ω = 0.835).

#### 4.2.5. Creative Self-Efficacy Scale (CSES) [42]

The 6-item CSES was used to evaluate one’s self-efficacy in creativity using a 5-point Likert scale ranging from 1 (never) to 5 (very often). Higher (average) scores indicate higher levels of creative self-efficacy. A sample item was “I trust my creative abilities”. The reliability of the CSES was satisfactory in the present study (α = 0.794; ω = 0.799).

### 4.3. Procedures

An online experiment was created using Qualtrics. An information sheet that showed the cover story of the study, data confidentiality, and participants’ rights was first presented to the participants. Participants who gave their consent were then allowed to proceed to the experiment and were randomly assigned to the experimental group or the control group. Then, all participants answered the CSES, followed by the writing task. After that, participants from both groups answered the AUT, MSPSS, and SPCS and were then debriefed about the main purpose of the study. The study was approved by the Scientific and Ethical Review Committee of the university (Ref: U/SERC/208/2020).

### 4.4. Analytical Plan

Intraclass correlation (ICC) based on mean-rating (k = 3), absolute agreement, 2-way mixed-effects mode, and their 95% confident interval (CI) was calculated to examine the inter-rater reliability in the ratings (for fluency, flexibility, and originality) among the three raters. After that, a multivariate analysis of covariance (MANCOVA) with group condition (experimental group vs. control group) as the independent variable was conducted to examine the effectiveness of the manipulation and the impact of the induced perceived social support on different measures of creativity (i.e., fluency, flexibility, originality, and self-rated creativity). We included creative self-efficacy as a covariate variable because the literature [43] has shown that creative self-efficacy is an antecedent factor of divergent thinking ability and self-reported creativity. Therefore, we (statistically) controlled for the effect of creative self-efficacy to clarify if perceived social support had a positive effect on creativity.

## 5. Results

### 5.1. Inter-Rater Reliability

The ICC results showed that the rating scores provided by the three evaluators were highly consistent with each other in the dimensions of divergent thinking: ICC = 0.980, 95% CI (0.973, 0.986) for fluency, ICC = 0.936, 95% CI (0.916, 0.953) for flexibility, and ICC = 0.965, 95% CI (0.953, 0.974) for originality. As a result, the three rating scores were averaged to indicate participants’ performance in each of the dimensions, respectively.

### 5.2. MANCOVA Results

Inspection of the Box’s test of equality of covariance matrices found that the assumption was not supported: Box’s M = 35.63, *F*(15, 70,912.346) = 2.28, *p* = 0.003. The multivariate tests indicated that the group effect was statistically significant: Pillai’s trace value = 0.239, *F*(5, 128) = 8.05, *p* < 0.001, partial η^2^ = 0.239. Finally, Levene’s test showed that the equality of error variances assumption was violated for the three divergent thinking indexes and self-rated creativity.

Tests of between-subject effects showed that a significant group difference was observed in all creativity measures (see Table 1). First, participants in the experimental group reported a higher perceived social support score than their counterparts in the control group, *F*(1, 132) = 8.75, *p* = 0.004, partial η^2^ = 0.062, supporting the effectiveness of the writing task in manipulating perceived social support. In addition, the induced perceived social support also enhanced participants’ creativity. Compared to those in the control group, participants who completed the writing task generated more responses, *F*(1, 132) = 7.04, *p* = 0.009, partial η^2^ = 0.051; diverse responses, *F*(1, 132) = 7.29, *p* = 0.008, partial η^2^ = 0.052; and more novel responses, *F*(1, 132) = 14.02, *p* < 0.001, partial η^2^ = 0.096 in the AUT. Similarly, the experimental group also rated themselves as more creative than the control group: *F*(1, 132) = 10.40, *p* = 0.002, partial η^2^ = 0.073.

### 5.3. Supplementary Analysis

As the assumptions of MANCOVA were violated, we log-transformed the three indexes of divergent thinking and the self-rated creativity score and submitted them to another MANCOVA to clarify if the results of the main analysis were confounded by the violation of assumptions. Both Box’s and Levene’s tests were not significant, indicating that the assumptions were supported. On the other hand, the multivariate test showed that the group effect was significant: Pillai’s Trace value = 0.172 (Wilks’ Lambda = 0.828), *F*(4, 129) = 6.70, *p* < 0.001, partial η^2^ = 0.172. In line with the main analysis, tests of between-subject effects showed that participants in the experimental group outperformed the counterparts in the control group in all measures of creativity: *F*(1, 132) = 4.60, *p* = 0.034, partial η^2^ = 0.034 for fluency; *F*(1, 132) = 5.07, *p* = 0.026, partial η^2^ = 0.037 for flexibility; *F*(1, 132) = 9.64, *p* = 0.002, partial η^2^ = 0.068 for originality; and *F*(1, 132) = 9.58, *p* = 0.002, partial η^2^ = 0.068 for self-rated creativity (see Appendix A for the mean and standard deviation values).

## 6. Discussion

The causal relationship between perceived social support and creativity (indexed by divergent thinking ability and self-rated creativity) was tested using an experimental design in the present study. Supporting our hypothesis, the results support that perceived social support leads to a higher level of creativity.

Several essential findings were found in the present study. First, in line with Xu and colleagues’ findings [37], our analysis results demonstrated that perceived social support was successfully manipulated by recalling three memories of receiving social support. The finding not only offers another piece of empirical evidence of the effectiveness of the writing task but also suggests a manipulation method for future studies to induce perceived social support temporarily in the laboratory.

The key finding of the present study was the positive effect of perceived social support on creativity. Our results extend the findings of correlational studies by shedding light on the causal relationship between the two constructs. Consistent with the literature [28,30], we found that the induced perceived social support consistently increased different measures of creativity. Individuals who perceived support from their social network not only generated more, diverse, and novel responses but also rated themselves as more creative. The congruency not only strengthens the plausibility of the findings but also highlights the robustness of the effect.

The present study contributes to the literature of social support in two ways. First, our results suggest a possible theoretical framework to account for the connections among social support, basic psychological needs, and creativity. Based on past findings of a positive relationship between the three constructs, respectively, [9,11,26] and our findings, future researchers are recommended to explore the indirect effect of social support on creativity via basic psychological needs. By integrating a social component with basic psychological needs, the socio-self-determination theory offers a more comprehensive understanding of creativity by emphasizing the essential role of socio-person fit. Second, our findings go beyond replicating the correlational relationship between social support and creativity and offer empirical support to the causal relationship of the variables. The latter has a high practical value in the promotion of creativity. For instance, future creativity training programs shall include a module to highlight the importance of social support and assist participants to expand their sources of social support.

Three limitations shall be taken into consideration when interpreting the results. First, the online experiment was conducted during the COVID-19 pandemic lockdown in Malaysia. The pandemic may influence participants perceived social support, while the online context restrained participants from clarifying their inquiries about the experiments. Therefore, it is important to replicate the present study physically when the pandemic is over and to control other confounding effects such as social-desirability bias. Second, the study was limited to undergraduate students in Malaysia. It is critical to verify the findings of the present study in other age populations and cultural contexts. In the same vein, the present study focused on the domain-general divergent thinking ability and creativity. It is intriguing to know whether the facilitative effect of perceived social support can also be observed in a specific domain of creativity. For instance, researchers may examine if perceived social support can promote individuals’ actual and self-rated artistic creativity using collage design [44] and the artistic creativity subscale of the 20-item Kaufman Domains of Creativity Scale [45]. Finally, the present study focused on the question of whether induced social support had an impact on creativity and did not explore the underlying mechanism. Future researchers, therefore, are recommended to investigate variables that may mediate the relationship between social support and creativity. A positive emotion such as happiness is a potential candidate because social support is positively related to happiness [19,20], which was found to enhance creativity [46]. In addition, as suggested by a reviewer, it is theoretically intriguing to examine the impact of lacking social support or negative social support on creativity by asking participants to recall and write down three memories of being socially unsupported.

## 7. Conclusions

Extending the past findings of the positive correlation between social support and creativity, the present study sheds light on the causality of the two constructs. The experimentally manipulated social support had a positive effect on different measures of creativity. Furthermore, our exploratory mediation analysis showed that social support had an indirect effect on self-reported creativity but not divergent thinking via creative self-efficacy. Taken together, social support is an essential factor in the development of creativity and shall be included in future creativity training and research.

## Figures and Tables

**Table 1 ijerph-19-11841-t001:** Mean and standard deviation of social support and creativity measures in the experimental and control groups.

		Experimental (*n* = 69)		Control (*n* = 66)	
No.	Variable	Mean	*SD*	Mean	*SD*
1	Social Support	5.67	0.68	5.31	0.66
2	Fluency	16.65	10.17	12.15	6.94
3	Flexibility	12.03	6.52	8.94	5.09
4	Originality	26.36	15.82	16.88	10.86
5	Self-rated creativity	3.75	0.46	3.48	0.36

Note. *n* = 135. All differences between the means in both groups are statistically significant at a 0.05 level.

## Data Availability

The datasets generated during the present study are available from the corresponding author upon request.

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
