# Peer review of "Perceived Social Support Increases Creativity: Experimental Evidence"

_ijerph, 2022, doi:10.3390/ijerph191811841_

Round 1

Reviewer 1 Report

I begin by congratulating the authors and researchers of the manuscript intitled “Perceived Social Support Increases Creativity: Experimental Evidence”. The topic is actual and very interesting.

My main concern is related to the “Manipulation of Perceived Social Support” in itself. As you describe, “participants in the experimental group were asked to recall and write down three memories of being socially supported, while the control group was required to write down three nearest objects around them”. But, maybe, you could add a third condition where participants would be asked to recall and write down three memories of being socially unsupported. This could be more interesting and could lead to a new vision of the lack of social support or the negative social support and its influence on creativity.

As a final comment, I call the authors' attention to self-citations. I counted 9 citations that could perhaps be reviewed and completed with other works besides those referred to as Tan, S-C.

For an article written in 6 pages, presenting 8 pages of references seems to me to be excessive. Also review the format of bibliographic references to correctly adapt to the IJERPH.

Good luck!

Reviewer 2 Report

Your paper presents an interesting examination of the influence of perceived social support and creativity among Malaysian college students. There are several aspects of your paper that need to be addressed in order for it to be suitable for publication. Your paper would benefit from being better situated in the social support literature. For example, your review of the literature overlooked seminal works in the area of social support (i.e. the work of Cohen and colleagues). Additionally, there is no mention in your paper about the source or type of social support that you are seeking to induce or that was induced in the manipulation (i.e. psychological, non-psychological - see Cohen & Wills 1985; Hobfoll 1988). This is an important theoretical oversight. While these issues may not be able to be completely resolved due to your study design they should be addressed and where they are unable to be resolved acknowledged as a limitation. Also, the discussion needs to be more strongly linked back to the literature. It is difficult to see what the theoretical or practical contribution is.

Round 2

Reviewer 2 Report

Thank you for addressing the review comments. Your paper is now placed more solidly in the social support literature. I am concerned about the percentage of self-citations. I understand the argument you have made but I believe that as your paper currently stands you are exceeding the boundary of what would be considered normal in our discipline. I would encourage you to ensure that all self-citations are actually needed. While I am loathed to give a prescribed percetagage, I would envisage that no more than 5-10% would seem appropriate. 

Author Response

The percentage of self-citation has been reduced from 2130% to 12.77% (i.e., 6 out of 47). The changes in references are highlighted in green color.